

# Growth parameters and responses of green algae across a gradient of phototrophic, mixotrophic and heterotrophic conditions

Erica B. Young[1,2], Lindsay Reed[1] and John A. Berges[1,2]

[1] Department of Biological Sciences, University of Wisconsin-Milwaukee, Milwaukee, Wisconsin, United States
[2] School of Freshwater Sciences, University of Wisconsin-Milwaukee, Milwaukee, Wisconsin, United States

Corresponding author
Erica B. Young, ebyoung@uwm.edu

## ABSTRACT

Many studies have shown that algal growth is enhanced by organic carbon and algal mixotrophy is relevant for physiology and commercial cultivation. Most studies have tested only a single organic carbon concentration and report different growth parameters which hampers comparisons and improvements to algal cultivation methodology. This study compared growth of green algae *Chlorella vulgaris* and *Chlamydomonas reinhardtii* across a gradient of photoautotrophic-mixotrophic-heterotrophic culture conditions, with five acetate concentrations. Culture growth rates and biomass achieved were compared using different methods of biomass estimation. Both species grew faster and produced the most biomass when supplied with moderate acetate concentrations ($1–4 \text{ g L}^{-1}$), but light was required to optimize growth rates, biomass yield, cell size and cell chlorophyll content. Higher acetate concentration ($10 \text{ g L}^{-1}$) inhibited algal production. The choice of growth parameter and method to estimate biomass (optical density (OD), chlorophyll *a* fluorescence, flow cytometry, cell counts) affected apparent responses to organic carbon, but use of OD at 600, 680 or 750 nm was consistent. There were apparent trade-offs among exponential growth rate, maximum biomass, and culture time spent in exponential phase. Different cell responses over $1–10 \text{ g L}^{-1}$ acetate highlight profound physiological acclimation across a gradient of mixotrophy. In both species, cell size vs cell chlorophyll relationships were more constrained in photoautotrophic and heterotrophic cultures, but under mixotrophy, and outside exponential growth phase, these relationships were more variable. This study provides insights into algal physiological responses to mixotrophy but also has practical implications for choosing parameters for monitoring commercial algal cultivation.

## INTRODUCTION

Growth of microalgal biomass for use in biotechnology, biofuel production, aquaculture, pharmaceutical applications and during wastewater treatment is of interest to the research community and algal biotechnology industry, so optimizing culture conditions for production efficiency is a major research focus (*Henley, 2019*). A range of different culture
approaches, including under strictly photoautotrophic as well as heterotrophic and mixotrophic conditions, are typically reported (*Wang, Yang & Wang, 2014*). Furthermore, a range of different methods and parameters for estimating algal growth are represented in published studies and applications (summarized in Table 1). However, inconsistent use of growth parameters makes comparisons difficult, hampering advances in algal cultivation methodology. A systematic comparison of algal growth estimates under different growth conditions is needed to determine which parameters are most robust and useful.

Optimizing microalgal growth has usually focused on the best combination of abiotic conditions, particularly light and inorganic macro- and micro-nutrient availability (*Smith & McBride, 2015*). But many studies over the last few decades have also shown that growth of some algal species can be enhanced by organic carbon additions to the culture medium (*Lee, 2001*; *Chen et al., 2011*). Some microalgal species can grow well mixotrophically and even heterotrophically, and mixotrophy may be a successful strategy for organisms in diverse natural habitats (*Burkholder, Glibert & Skelton, 2008*; *Selosse, Charpin & Not, 2017*). Energy use flexibility in algae involves biochemical interactions between photosynthetic light harvesting and carbon fixation and respiratory organic carbon processing in response to light and organic carbon (C) supply (*Liu et al., 2009*; *Roach, Sedoud & Krieger-Liszkay, 2013*; *Xie et al., 2016*; *Li et al., 2020*). There is profound dynamic physiological acclimation required to balance autotrophic and heterotrophic carbon use (*Heifetz et al., 2000*; *Bogaert et al., 2019*).

Considerable attention has been focused on optimizing algal cell growth rates using organic C supplements, or even exclusive heterotrophic cultivation (*Lee, 2001*; *Bogaert et al., 2019*; *Pang et al., 2019*). Support of algal growth by addition of organic C sources can help alleviate light limitation in high density cultures, reduce light requirements, and algae may be able to use organic C sources available as waste products from other processes, possibly improving biomass production efficiency and costs (*Chen et al., 2011*; *Nirmalakhandan et al., 2019*; *Ummalyma et al., 2022*). Relatively few algal taxa have been examined in laboratory or commercial cultivation (*Luo et al., 2017*; *Nirmalakhandan et al., 2019*; *Wang, He & Young, 2020*), although green algae which grow fast and tolerate high nutrient concentrations include *Chlorella*, *Chlamydomonas* and *Scenedesmus* species, for which there is also information about growth, physiology and genetics important for researchers. While some taxa can grow heterotrophically, growth rates are often lower than with light, and many cells may need light to optimize production of economically-important metabolites such as lipids, proteins or pigments (*Lee, 2001*; *Chen et al., 2011*; *Karimian, Mahdavi & Gheshlaghi, 2022*) and mixotrophy may yield superior biomass production than solely heterotrophic or photoautotrophic conditions (*Li et al., 2020*). Several organic C sources have been applied, most commonly acetate or glucose, glycerol but also amino acids or organic hydrolysates or chemical by-products (*Chen et al., 2011*; *Cheng et al., 2022*). Studies have compared growth and yield across different taxa, culture type, inorganic and organic C sources and supply conditions, but typically using only one organic C concentration (Table 1). To understand how cells acclimate photosynthetic physiology to respond to autotrophic, mixotrophic and heterotrophic conditions, and to optimize cell culture production for applied purposes, we need to

**Table 1 Summary of growth parameters reported for studies comparing mixotrophy (MX), heterotrophy (HT) and photoautotrophy (PA) with an emphasis on green algal species with some other algal groups represented.**

| Phylum | Species | Trophic modes tested | Culture type | Organic C used | HT growth | Growth effect | Growth parameters assessed | Other effects | Reference |
|---|---|---|---|---|---|---|---|---|---|
| Chlorophyta | *Asterarcys* sp. SCS-1881 | PA, MX, HT | batch | gluc | no | MX>PA, HT | OD750, counts, mass | MX>PA protein synthesis, MX<PA pigment, TAG | *Li et al. (2020)* |
| Chlorophyta | *Chlamydomonas acidophila* | PA, MX | semi-cont | gluc | nd | MX>PA | volume | cells could use DOC but not POC sources | *Tittel et al. (2005)* |
| Chlorophyta | *Chlamydomonas acidophila* | PA, MX, HT | semi-cont | gluc | no | MX>PA>>HT | OD800 | MX>PA Pmax, cell size; PA>MX Chl | *Spijkerman, Lukas & Wacker (2017)* |
| Chlorophyta | *Chlamydomonas acidophila* | PA, MX, HT | batch | gluc, acet | v low | PA>MX>HT | counts | acet toxic in MX; cell size variable in PA, MX | *Souza et al. (2017)* |
| Chlorophyta | *Chlamydomonas humicola* | PA, MX, HT | batch | acet | yes | MX>HT>PA | counts, OD680 | MX>PA, HT biomass, protein, chl; PA>MX, HT lipid | *Laliberte & De la Noué (1993)* |
| Chlorophyta | *Chlamydomonas reinhardtii* | PA, MX, HT | batch | acet grad | yes | no effect | OD750 | PS reduced with incr. [acet] | *Heifetz et al. (2000)* |
| Chlorophyta | *Chlamydomonas reinhardtii* | MX | batch | acet grad | nd | μ>mod acet | counts | biomass, starch, protein incr with [acet]. | *Bogaert et al. (2019)* |
| Chlorophyta | *Chlamydomonas reinhardtii* | MX, HT | batch | acet | yes | MX>HT | counts | TAG production: MX>HT | *Singh et al. (2014)* |
| Chlorophyta | *Chlorella protothecoides* | PA, HT | split MX | gluc, acet | yes | MX>HT>PA | OD750, mass | Mx>PA lipid yield after N deprivation | *Sim et al. (2019)* |
| Chlorophyta | *Chlorella protothecoides* | MX, HT | batch | gluc | yes | PA>HT | counts, mass | HT>PA lipid, same yield at stat phase | *Rosenberg et al. (2014)* |
| Chlorophyta | *Chlorella pyrenoidosa* | MX, HT | batch | gluc | yes | MX>HT | mass | MX>HT NH4+ removal | *Wang et al. (2021)* |
| Chlorophyta | *Chlorella pyrenoidosa* | MX HT | batch | gluc | yes | MX>HT | mass | MX>HT N removal | *Cheng et al. (2022)* |
| Chlorophyta | *Chlorella sorokiniana* | MX, HT | batch | mal | no | MX>PA>>HT | OD550 | mal used only in light; PA>MX Rubisco act | *Qiao, Wang & Zhang (2009)* |
| Chlorophyta | *Chlorella sorokiniana* | PA, MX | airlift batch | acet | nd | MX>PA | counts, mass | MX cells retain PS capacity | *Cecchin et al. (2018)* |
| Chlorophyta | *Chlorella sorokiniana* | PA, MX | batch | acet | nd | MX>PA | OD750 | acet reduces photoinhibition | *Xie et al. (2016)* |
| Chlorophyta | *Chlorella sorokiniana* | HT | batch | acet, butyr, lact | yes | acet incr, butyr inhib | OD800, mass | acet:butyr ratio affects growth | *Turon et al. (2015)* |

(Continued)

| Phylum | Species | Trophic modes tested | Culture type | Organic C used | HT growth | Growth effect | Growth parameters assessed | Other effects | Reference |
|---|---|---|---|---|---|---|---|---|---|
| Chlorophyta | *Chlorella sorokiniana* | PA, MX, HT | batch | gluc | yes | HT>MX>PA | counts | HT>PA lipid | *Rosenberg et al. (2014)* |
| Chlorophyta | *Chlorella sorokiniana* | PA, MX, HT | batch | gluc, acet, glyc | yes | MX>PA,HT | OD680, mass | HT>MX protein, MX>PA, HT lipids | *Karimian, Mahdavi & Gheshlaghi (2022)* |
| Chlorophyta | *Chlorella vulgaris* | PA, HT | batch | gluc | yes | PA>HT | counts | HT>PA lipid | *Rosenberg et al. (2014)* |
| Chlorophyta | *Chlorella vulgaris* | PA, MX, HT | semi-cont | gluc | yes | MX>PA=HT | OD800 | MX larger cells; diff FA HT-PA; HT>PA C:P ratio | *Spijkerman, Lukas & Wacker (2017)* |
| Chlorophyta | *Chlorella vulgaris* | PA, MX, HT | column, panel | gluc | no | MX>PA | OD750, mass | needs light to use gluc | *Subramanian, Yadav & Sen (2016)* |
| Chlorophyta | *Chlorella vulgaris* | MX | batch micropl | acet, gluc | nd | glu+acet>acet | OD750, FC, biomass | MX *vs* PA changes OD/ FC-mass relationship | *Chioccioli, Hankamer & Ross (2014)* |
| Chlorophyta | *Chlorella sp.* HS2 | HT, MX | batch, multich | gluc, yeast | yes | MX>HT | counts, mass | MX>HT pigment | *Kim et al. (2020)* |
| Chlorophyta | *Chlorella* | PA, MX, HT | batch | gluc 2% | yes | MX>PA>HT | OD750, mass | PA>MX, HT lipid accum | *Ratha et al. (2013)* |
| Chlorophyta | *Dunaliella bardawil* | PA, MX, HT | batch | acet, gluc | v low | MX>PA>>HT | counts | gluc>acet higher β-carotene, lipid | *Chavoshi & Shariati (2019)* |
| Chlorophyta | *Graesiella sp.* | PA, MX, HT | batch | gluc | v low | MX>HT, PA | mass | PSII act lost under MX, HT | *Zili et al. (2017)* |
| Chlorophyta | *Scenedesmus obliquus* | PA, MX, HT | batch | yeast, Bold | yes | MX> PA, HT | counts | MX>PA, HT SOD act | *Pokora, Aksmann & Tukaj (2011)* |
| Chlorophyta | *Scenedesmus obliquus* | PA, MX, HT | batch | acet | yes | MX>PA>HT | OD680, counts | isocitrate lyase act with acet | *Combres et al. (1994)* |
| Chlorophyta | *Scenedesmus obliquus* | PA, MX, HT | batch | acet | yes | MX> PA>HT | mass | MX>PA, HT N, P removal, lipid content | *Choi et al. (2019)* |
| Chlorophyta | *Scenedesmus obliquus* | MX, HT | batch, matrix | acet | nd | nd | OD682 | MX>HT N removal | *Liu et al. (2019)* |
| Chlorophyta | *Scenedesmus obliquus* | PA, MX | batch | acet, pyrv | nd | MX>PA | mass | Pyrv/acet diff effects on growth, cell parameters | *Mansouri et al. (2022)* |
| Chlorophyta | *Scenedesmus sp.* | PA, MX, HT | batch | molas | yes | MX>HT>PA | counts, mass | PS, chl maintained with molas | *Kamalanathan et al. (2017)* |
| Chlorophyta | *Scenedesmus* | PA, MX, HT | batch | molas | yes | HT>PA | counts, mass | HT>PA biomass, μ, lipid | *Kamalanathan et al. (2018)* |

| Phylum | Species | Trophic modes tested | Culture type | Organic C used | HT growth | Growth effect | Growth parameters assessed | Other effects | Reference |
|---|---|---|---|---|---|---|---|---|---|
| Chlorophyta | *Chlorococcum* | PA, MX, HT | batch | gluc 2% | yes | MX>PA>HT | OD750, mass | PA>MX, HT lipid accum | *Ratha et al. (2013)* |
| Haptophyta | *Isochrysis galbana* | PA, MX, HT | batch | glyc | no | MX>PA>>HT | counts, mass | >biomass with glyc 25–50 mM | *Alkhamis & Qin (2013)* |
| Bacillariophyta | *Pavlova lutheri* | PA, MX, HT | batch | gluc, glyc, acet, sucr | v low | MX>HT | counts, OD750 | Sucr>gluc>acet growth | *Bashir et al. (2019)* |
| Bacillariophyta | *Phaeodactylum tricornutum* | PA, MX | batch | glyc | nd | MX>PA | counts | N deprivation -> lipid accum | *Villanova et al. (2017)* |
| Rhodophyta | *Galdieria sulphuraria* | PA, MX, HT | batch | sorb | yes | MX>PA+HT | counts, mass | org C stimulates PS *via* $CO_2$ supply | *Curien et al. (2021)* |
| Rhodophyta | *Galdieria sulphuraria* | PA, MX | batch 700 L | ww | no | nd | OD750, mass | low pH cultivation suppressed bacteria | *Nirmalakhandan et al. (2019)* |
| Rhodophyta | *Galdieria sulphuraria* | PA, MX | batch/chemo | Gluc | yes | MX>PA | mass | MX>PA pigment prod | *Abiusi et al. (2022)* |
| Ochryophyta | *Nannochloropsis oceanica* | PA, MX | batch | acet | ND | MX>PA | OD750 | TCA, C4 cycle stim by DIC +DOC | *Li et al. (2020)* |
| Cyanobacteria | *Nostoc* | PA, MX, HT | batch | gluc 2% | yes | MX>PA, HT | OD750, mass | MX>PA, HT lipid accum | *Ratha et al. (2013)* |
| Cyanobacteria | *Phormidium, Anabaena* | PA, MX, HT | batch | gluc 2% | yes | MX>PA, HT | OD750, mass | MX>PA, HT lipid accum | *Ratha et al. (2013)* |

**Notes:**
PA, photoautotrophy; MX, mixotrophy; HT, heterotrophy.
nd, not determined; Chl, chlorophyll; OD, optical density; counts, counts of cell by microscopy or automated counter unless; FC, flow cytometry specified; TAG, triacylglycerol; FA, fatty acids; incr, increasing; μ, growth rate.
Culture types: semi-cont, semi-continuous culture; panel, panel bioreactor; micropl, microplate; multich, multichannel bioreactor; matrix, matrix immobilized; chemo, chemostat.
Organic C substrates: gluc, glucose; acet, acetate; butyr, byturate; fruct, fructose; glyc, glycerol; lact, lactate; mal, malate; meat, meat extract; molas, molasses; pep, peptone; pyrv, pyruvate; sucr, sucrose; sorb, sorbitol; yeast, yeast extract; Bold, Bolds basal medium.
grad, gradient; ww, wastewater; act, activity; DIC, dissolved inorganic C; DOC, dissolved organic C.

characterize cell growth and parameters such as size and pigment content over a gradient of organic C concentrations.

For applied biotechnological cultivation of microalgal species, there are several useful culture parameters to compare. Growth rate is important, but also the maximum biomass, or carrying capacity, of the culture conditions may be an important parameter for commercial application, as well as how long cultures can sustain maximum growth rates in culture (*Andersen, Faeerovig & Hessen, 2007*; *Smith & McBride, 2015*), but most studies do not report these parameters. In assessing culture growth responses, direct mass measurements require relatively dense and/or large culture volumes for accuracy (*Subramanian, Yadav & Sen, 2016*), so many applied studies employ rapid algal biomass

assessment methods such as optical density (OD) at 600 or 750 nm, which is easy and relatively cheap to measure (*Chioccioli, Hankamer & Ross, 2014*). Other studies have used 680 nm to target absorbance by algal chlorophyll (*Xiao et al., 2015*; *Luo et al., 2020*). Counts of cells using manual microscopy counts or more automated methods are common, often used in combination with other parameters (*Chioccioli, Hankamer & Ross, 2014*; *Kamalanathan et al., 2018*). Chl-based estimates of biomass, typically based on chlorophyll *a* fluorescence, require specialized fluorometers, but are sensitive and have been widely applied to estimate biomass and physiological parameters in field and culture studies (*Kolber & Falkowski, 1994*; *Young & Beardall, 2003*; *Chen et al., 2017*). These parameters target different cell characteristics so it is unclear if using different growth parameters will result in similar or different growth rates or conclusions in responses to photoautotrophic, mixotrophic or heterotrophic conditions. More detailed comparisons of different growth parameters for algal growth applications are needed. Furthermore, depending on the application for rapid *vs.* dense biomass production, different parameters, including cell density, size and pigment content may be most useful. Specific comparisons of these measuring parameters have not often been carried out across different growth conditions.

To address the need for more detailed comparisons of cell culture parameters, and responses of cells over a gradient of mixotrophic, autotrophic and heterotrophic conditions, this study used two species of green algae commonly used in biotechnology and for biofuels production. The study focused on **three research aims** and related hypotheses:

1. To compare the use of different parameters for estimating growth and biomass production under photoautotrophic, mixotrophic and heterotrophic growth conditions in two green algal species. We hypothesize that growth rates will vary with mode of energy supply and that different biomass estimating parameters will result in different growth rate relationships.
2. To examine additional important algal culture parameters, *i.e.*, maximum biomass achieved and time spent in exponential phase of batch cultures, across a gradient of photoautotrophic, mixotrophic and heterotrophic growth conditions. We hypothesize that there will be a trade-off between maximum exponential growth rate and maximum biomass achieved and culture time spent in exponential phase.
3. To examine algal cell size and cell chlorophyll responses to photoautotrophic, mixotrophic and heterotrophic culture conditions over the culture growth cycle. We hypothesize that cell size and chlorophyll per cell will change with trophic energy mode and culture growth phase.

# MATERIALS AND METHODS

## Cultures

*Chlorella vulgaris* (UTEX 259) was obtained from UTEX (utex.org) and *Chlamydomonas reinhardtii* (c-9 wt) was obtained from *Chlamydomonas* resource center

(chlamycollection.org/). Cultures of each species were maintained in stock cultures and used to inoculate cultures in one liter flasks with 500 mL of DY-V medium (*Anderson et al., 2005*), all maintained at 18 °C. Magnetic stir bars and filter-sterilized (Whatman GF/F) air agitated the medium, and flasks were fitted with a sampling port with a syringe. Irradiance was supplied by fluorescence light tubes (Philips Alto TL841 HO) at ~30 μmol $m^{-2}$ $s^{-1}$ at the flask surface with L:D periodicity of 14:10 h. For mixotrophic and heterotrophic conditions, organic C was added as sterile-filtered (0.2 μm) sodium acetate (Fisher Scientific, Hanover Park, IL, USA) which is most often included as a C source in synthetic wastewater (*Xiao et al., 2015*; *Luo et al., 2020*). For mixotrophic cultures, acetate from a 100 g $L^{-1}$ stock solution was added to cultures for final concentrations of 1, 2, 3, 4, 10 g $L^{-1}$ in the growth medium. Heterotrophic cultures were supplied with 2 g $L^{-1}$ sodium acetate and culture flasks were covered with black felt to exclude all light and placed in the same growth chamber at 18 °C. Four replicate cultures were used for each treatment and species. Changes in algal biomass were evaluated by changes in chlorophyll *a* (chl) fluorescence and optical density (OD) measured in culture subsamples withdrawn from the sampling port within a sterile hood. Chl fluorescence was measured twice daily in a 4 mL subsample using a benchtop fluorometer (TD-700; Turner Designs, San Jose, CA, USA). OD of the same sample was also measured at 600, 680 and 750 nm daily in 1 cm cuvette in a spectrophotometer (DU-640; Beckman Coulter, Brea, CA, USA).

## Growth parameters

Cultures were monitored for ~6–8 days covering lag, exponential and into stationary phase. OD and chl fluorescence data were collected and plotted over time, then natural log transformed to determine exponential growth regions for growth rate ($d^{-1}$) calculations from beginning (initial) to end (final) of the exponential period using specific growth rate equation (μ) (*Fogg & Thake, 1987*):

$$\mu = \ln(\text{biomass}_{\text{final}}) - \ln(\text{biomass}_{\text{initial}})/\text{time}_{\text{final}} - \text{time}_{\text{initial}}$$

The length of the exponential growth period, from end of lag phase to beginning of stationary phase, was measured from the linear portion of the natural log plots. Maximum biomass achieved at stationary phase (as a measure of carrying capacity, K) was also calculated using a linear regression through at least three points in the stationary phase of the culture where the maximum chl fluorescence or OD values were observed for each culture. The parameters compared from the growth curves are shown in Fig. 1.

To extend the comparison of biomass parameters and examine the possibilities of OD increases related to cell size, differences in chl per cell, or biomass increases related to growth of bacteria rather than algal cells, especially in mixotrophic and heterotrophic cultures, an additional experiment compared growth rates based on measurements of chl fluorescence, OD (as above) but also included counts of algal cells but comparing the 2 g $L^{-1}$ acetate concentration for mixotrophic and heterotrophic conditions. Cells were counted using microscopy with a haemocytometer of samples collected daily and

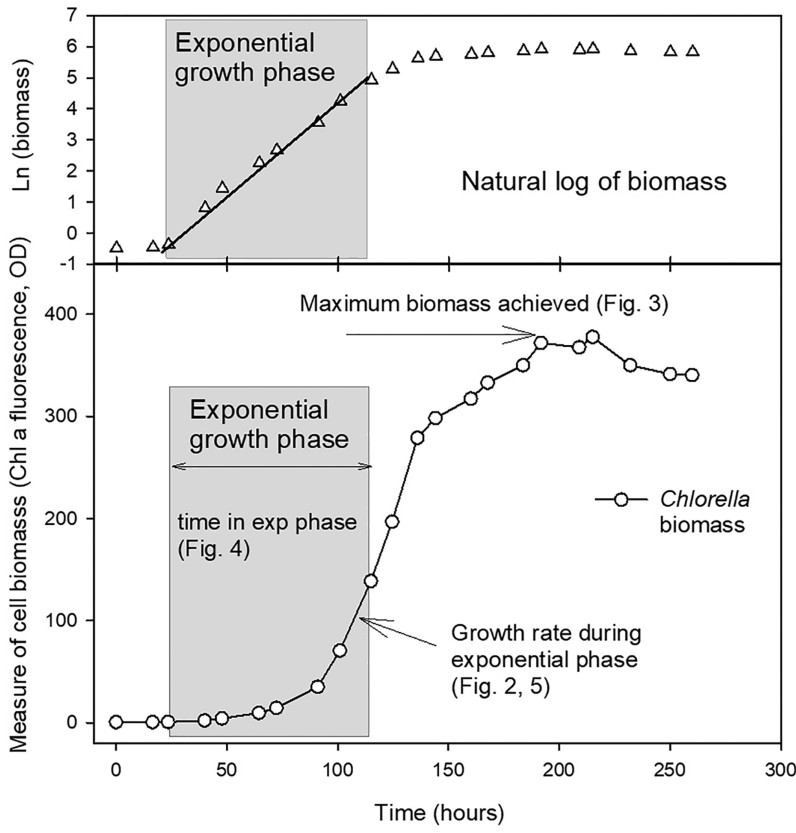

**Figure 1 Example growth curve showing growth parameters used for analysis.** Plot of growth of *Chlorella vulgaris* in a batch culture example to illustrate the growth parameters reported in Figs. 2–5. The period of exponential growth phase (grey box) was estimated from the linear portion of a plot of the natural log of biomass over time (top plot).

preserved with acid Lugol's iodine reagent and stored in the dark at 4 °C. Cells were also counted and cell size and chl fluorescence in algal cell populations examined by flow cytometry (FC). Three treatments were compared: photoautotrophic, mixotrophic with 2 g L$^{-1}$ acetate and heterotrophic with 2 g L$^{-1}$ acetate conditions, as described above, with three replicate cultures of each treatment. FC was carried out on a benchtop flow cytometer (BD Accuri C6, 488 nm blue laser), using forward scatter (FSC-H) as a proxy for cell size and chl fluorescence signal (FL-3-H) as an estimate of per cell chl fluorescence, both within the previously-defined gates for each species, using peak height viewed on a histogram. Gating of each algal species was performed separately and used for total cell counts. Typical FC parameters were a 10 μm core size, a 14 μL min$^{-1}$ flow rate and a threshold of 800 units on FL-3. FC plots were analyzed, and counts were collected using BD Accuri C6 system software.

## Analysis

The average of parameters (growth rate, time spent in exponential phase, maximum cell density), were calculated from replicate cultures for each treatment and species. Culture condition treatments were compared with a one-way ANOVA or a two-way ANOVA with

treatment and measured parameter as factors, separately for each species, and to compare culture treatment and species as factors. Calculated parameters using each of the three wavelengths for OD (600, 680, 750 nm) were compared with two-way ANOVA of culture treatment and wavelength, but no differences were found, so OD at 680 nm only was used for all further statistical treatment comparisons. Within each species, treatment effects on each parameter were compared with one-way ANOVA with Tukey's or Holm-Sidak multiple comparisons. The data for time spent in exponential phase failed equal variance test for ANOVA so a Kruskal–Wallis non-parametric test on ranks was performed, with Tukey pairwise comparisons. For experiments including FC and haemocytometer counts, along with chl fluorescence and OD at 680 nm, growth rates were calculated on the same time periods across all four measuring methods and compared with two-way ANOVA using treatment and growth rate estimation method as factors, separately for each species, with Holm-Sidak pairwise comparisons. All statistical analyses were made using Sigmaplot (v. 12.5; Systat Software Inc, San Jose, CA, USA).

# RESULTS

## Exponential growth rates

Both species grew rapidly in the batch cultures with similar ranges of exponential growth rates over photoautotrophic (0 g L$^{-1}$), mixotrophic and heterotrophic conditions (Fig. 2). Based on chl fluorescence, *Chlorella vulgaris* grew at similar rates for all light-grown cultures (maximum 1.17 ± 0.17 d$^{-1}$) but the growth rate in heterotrophic cultures was significantly lower than in other treatments (Fig. 2A, one-way ANOVA with Holm-Sidak multiple comparisons, $p = 0.007$, F = 4.11, df = 6, $n = 4$). *Chlamydomonas reinhardtii* grew faster (maximum 1.54 ± 0.35 d$^{-1}$) with 2 g L$^{-1}$ than other conditions (one-way ANOVA, $p < 0.001$, F = 18.7, df = 6, $n = 4$) (Fig. 2B), and growth rate in mixotrophic 10 g L$^{-1}$ and heterotrophic conditions was slower than mixotrophic cultures supplied with 2, 3 and 4 g L$^{-1}$.

Growth of cells in different treatments based on changes in chl fluorescence *vs.* optical density (OD) yielded similar growth rates for light-grown cultures and between the two species; OD-based growth rates excluding heterotrophic cultures for *C. vulgaris* were 1.11 ± 0.060 d$^{-1}$ and for *C. reinhardtii* 0.92 ± 0.042 d$^{-1}$ (Figs. 2C, 2D). However, OD-based growth rates of heterotrophic cultures were significantly higher than the light-grown cultures of both species (one-way ANOVA with Holm-Sidak multiple comparisons: *C. vulgaris* $p < 0.001$, df = 6, F = 57.496; *C. reinhardtii* $p = 0.023$, ANOVA, df = 6, F = 17.26), in contrast to lower culture growth rates in heterotrophic conditions when based on chl fluorescence (Fig. 2). Within OD wavelengths, there were no significant differences between growth rates determined using OD at 600, 680 or 750 nm (two-way ANOVA). Based on OD at 680 nm, all photoautotrophic and mixotrophic cultures of *C. vulgaris* grew at similar rates but there was some variability in growth rates among treatments in *C. reinhardtii* (Figs. 2C, 2D).

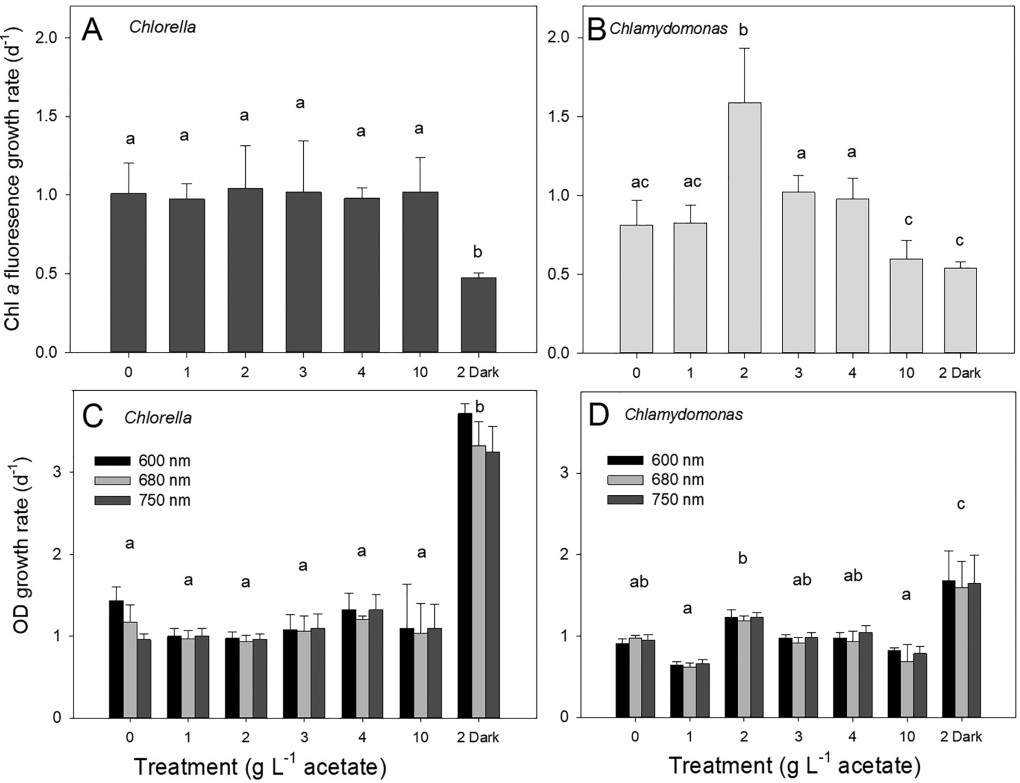

**Figure 2 Algal growth rates based on chl a fluorescence and optical density measurements.** Growth rates of *Chlorella vulgaris* and *Chlamydomonas reinhardtii*, based on Chl *a* fluorescence (A and B) or OD (600, 680, 750 nm; C and D). Growth rates were calculated during exponential growth phase of batch cultures. Cultures were maintained in light with 0 g L$^{-1}$ acetate (photoautotrophic conditions), supplied with light and acetate (1, 2, 3, 4 or 10 g L$^{-1}$) (mixotrophic), or 2 g L$^{-1}$ acetate in the dark (heterotrophic). Bars are mean + standard deviation from four replicate cultures, and treatments with significant different values are indicated with different lowercase letters (1-way ANOVA). For OD, statistical differences between treatments are shown only for OD at 680 nm.

## Maximum biomass achieved

The maximum biomass achieved during stationary phase in cultures of *C. vulgaris* based on chl fluorescence was 303 ± 91 relative fluorescence units (rfu) in photoautotrophic cultures, which was similar to maximum biomass for mixotrophic cultures supplied with 1, 2 and 4 g L$^{-1}$ acetate but higher than biomass for 3 and 10 g L$^{-1}$ and heterotrophic cultures (Fig. 3A; ANOVA with Holm-Sidak multiple comparisons, $p = 0.002$, F = 20.5, df = 6, $n = 4$). For *C. reinhardtii*, the maximum chl fluorescence was also highest in photoautotrophic cultures with 376 ± 59 rfu, and maximum chl fluorescence was significantly lower in 10 g L$^{-1}$ and heterotrophic cultures than the other treatments (Fig. 3B) (ANOVA with Holm-Sidak multiple comparisons, $p < 0.001$, F = 31.79, df = 6, $n = 4$).

There were no treatment effect differences in maximum biomass based on use of different OD wavelengths (600, 680, 750 nm) (two-way ANOVA with wavelength and treatment as factors). *C. vulgaris* cultures only achieved higher maximum values than *C. reinhardtii* at 1 g L$^{-1}$ acetate (Figs. 3B, 3C) (two-way ANOVA with species and

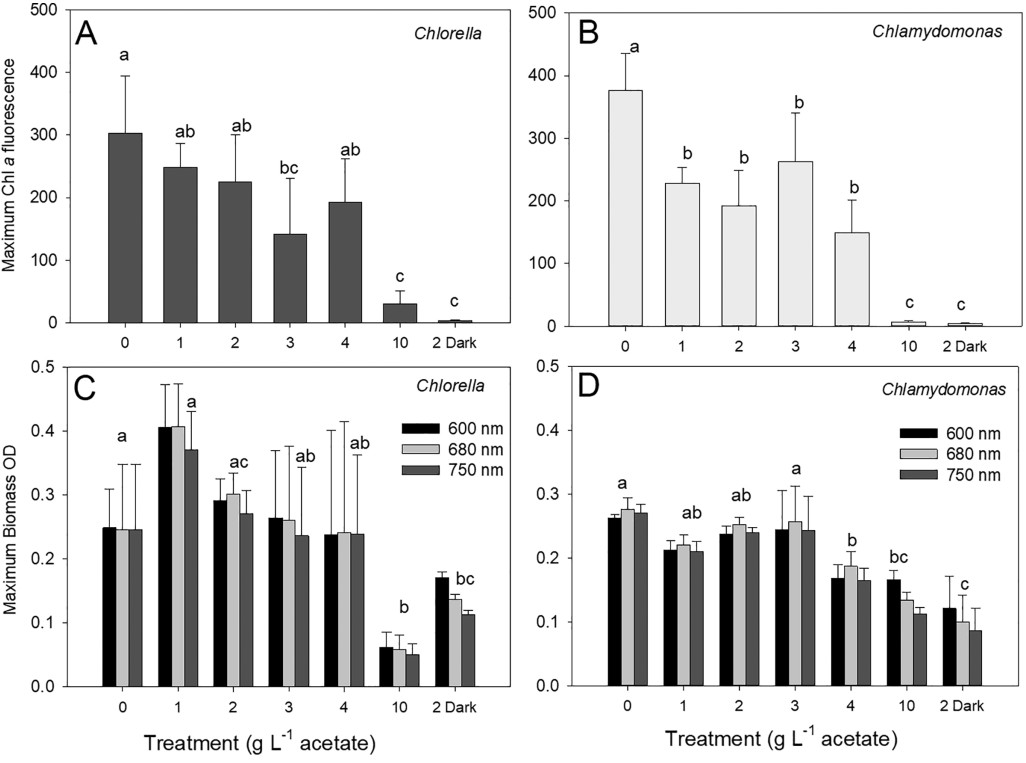

**Figure 3 Maximum algal biomass achieved in cultures.** Maximum biomass achieved in batch cultures of green algae, based on Chl a fluorescence (A and B) or OD (600, 680, 750 nm; C and D). Values for Chl *a* fluorescence or OD were estimated during stationary phase of growth. Treatments are the same as Fig. 2. Bars are mean + standard deviation from four replicate cultures, and treatments with significant different values are indicated with different lowercase letters (one-way ANOVA). For OD, statistical differences are shown for OD at 680 nm.

treatment as factors, Holm-Sidak multiple comparisons, $p < 0.001$, F = 11.5, df = 6, $n = 4$). The highest *C. vulgaris* maximum biomass in mixotrophic conditions was found with 1 g L$^{-1}$ acetate (0.408 ± 0.07 OD) but similar biomass OD were observed for photoautotrophic and mixotrophic cultures except 10 g L$^{-1}$ acetate (Fig. 3C). Heterotrophic and mixotrophic *C. vulgaris* cultures supplied with 10 g L$^{-1}$ acetate achieved lower maximum OD than all other treatments (ANOVA, $p = 0.011$, F = 16.56, df = 6, $n = 4$). Maximum OD of *C. reinhardtii* cultures was highest in photoautotrophic cultures (0.276 ± 0.02 OD) and mixotrophic cultures supplied with 1–3 g L$^{-1}$ acetate (Fig. 3D) and maximum biomass was significantly lower in heterotrophic cultures than all treatments except 10 g L$^{-1}$ mixotrophic (ANOVA, $p < 0.001$, F = 22.73, df = 6, $n = 4$).

## Culture time in exponential phase

Discrete culture sampling times resulted in unequal variance in time in exponential phase values (some treatments showed no variation), so comparisons were made with Kruskal–Wallis ANOVA on ranks. There were no significant differences in times in exponential phase between the two species, for either parameter. The longest times were in photoautotrophic and mixotrophic cultures with exponential phase of 3–5 days and the

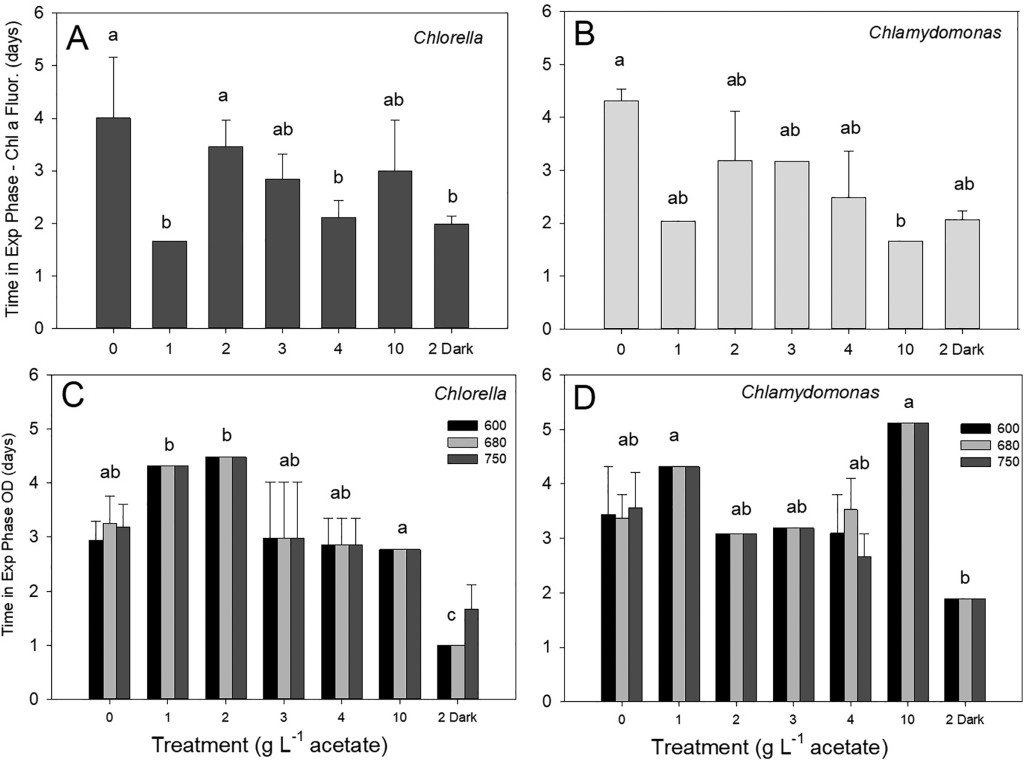

**Figure 4 (A–D) Culture time period spent in exponential phase of growth.** Time spent in exponential growth phase for batch cultures of green algae based on Chl *a* fluorescence or OD (at 600, 680 or 750 nm). Treatments are as in Fig. 2. Bars are mean + standard deviation from four replicate cultures, and treatments with significant different values are indicated with different lowercase letters (one-way ANOVA). For OD, statistical differences are shown for OD at 680 nm only.

shortest times in exponential phase were in heterotrophic or higher acetate concentration mixotrophic treatments (1–2 days). The maximum culture time in exponential phase for *C. vulgaris* based on chl fluorescence was highest in photoautotrophic cultures (4.0 ± 1.2 d), but with similar times across heterotrophic cultures and most mixotrophic cultures (Fig. 4A). For *C. reinhardtii*, maximum time in exponential phase was also observed in photoautotrophic cultures (103.2 ± 5.28 h (4.3 ± 0.22 days)) but with similar times across heterotrophic cultures and most mixotrophic cultures (Fig. 4B). When based on OD (680 nm), the shortest time in exponential phase was also in the *C. vulgaris* heterotrophic cultures (<48 h) (K-W ANOVA on Ranks, $p = 0.001$, H = 22.29, df = 6, $n = 4$) and in *C. reinhardtii*, heterotrophic cultures grew in exponential phase for shorter time than in 1 and 10 g L$^{-1}$ treatments (K-W ANOVA on Ranks, $p = 0.001$, H = 22.68, df = 6, $n = 4$) (Figs. 4C, 4D). There was no indication that a longer exponential phase led to higher maximum biomass, although for *C. reinhardtii* cultures grown in the light there was apparently a slight negative relationship (Fig. S1).

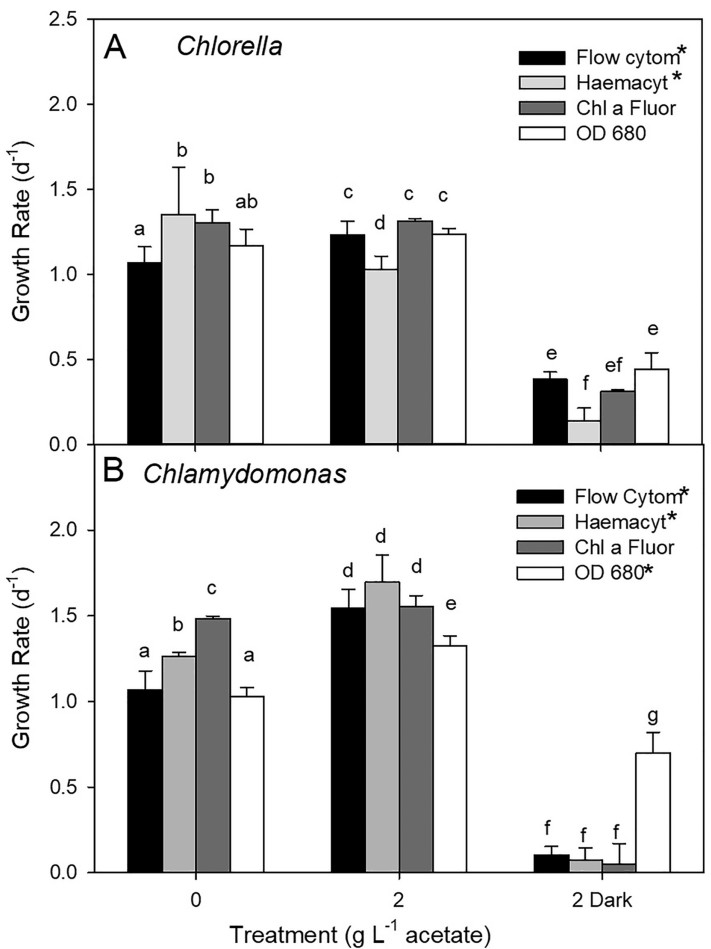

**Figure 5 Algal culture growth rates based on different estimates of biomass.** (A) *Chlorella*. (B) *Chlamydomonas*. Growth rates calculated for batch cultures of green algae based on cell counts from flow cytometry and haemocytometer counts, chl fluorescence and OD during exponential phase of growth. Three treatments were 0 g L$^{-1}$ acetate (photoautotrophic), 2 g L$^{-1}$ acetate plus light (mixotrophic), and 2 g L$^{-1}$ acetate in the dark (heterotrophic). Bars are mean + standard deviation from three replicate cultures. Statistically significant growth rates within each species and treatment are indicated with different lowercase letters (two-way ANOVA).

## Measuring methods for growth rate estimates

Growth rates comparisons between estimates based on cell counts using flow cytometry (FC), cell counts from microscopy, and culture density based on chl fluorescence and OD for three treatments showed some differences (two-way ANOVA with treatment and estimate parameter as factors and Holm-Sidak multiple comparisons) although the highest growth rate estimates were not consistently found with one estimation method (Fig. 5). In photoautotrophic and mixotrophic cultures, the highest rates were based on microscopy counts and chl fluorescence. In heterotrophic cultures, (OD clearly gave higher growth rate estimates for *C. reinhardtii* n = 3) but in *C. vulgaris*, OD based growth rates were only higher than microscopy counts (two-way ANOVA, $p < 0.001$, F = 350.9, df = 2). With all methods, growth rate was slowest in heterotrophic cultures of *C. vulgaris*, and with all methods except OD for *C. reinhardtii* (ANOVA treatment effect $p < 0.001$ for both

species). In *C. vulgaris*, growth rates estimated by FC were highest in mixotrophic (2 g L$^{-1}$ acetate) and lowest in heterotrophic cultures (Fig. 5A, two-way ANOVA, $p < 0.001$, $F = 114.5$, df = 2, $n = 3$), but photoautotrophic and mixotrophic cultures showed similar growth rates when estimated with chl fluorescence or OD. With haemacyometer microscopy counts, *C. vulgaris* growth rates were highest in photoautotrophic cultures (ANOVA, $p < 0.001$, $F = 114.5$, df = 2, $n = 3$). In *C. reinhardtii*, growth rates were also highest in mixotrophic culture and lowest in heterotrophic cultures with FC, microscopy counts and OD but with chl fluorescence, growth rates were similar growth in photoautotrophic and mixotrophic cultures (Fig. 5B).

### Cell size and fluorescence

Cell size and chl fluorescence per cell varied over the culture growth phases with distinct patterns between the three growth conditions (Fig. 6). In both species, in photoautotrophic and mixotrophic conditions, there was an initial increase in cell size but during stationary phase, cell size declined in photoautotrophic cells and but increased markedly in mixotrophic cells (Figs. 6C, 6D). In contrast, in heterotrophic cultures, cell size in both species declined until the start of stationary phases then increased in *C. vulgaris* but not in *C. reinhardtii* (Figs. 6E, 6F). Chl fluorescence per cell showed some similar trends to cell size, but some differences suggest that chl fluorescence per cell was not solely based on cell size. The closest correlation between mean cell size and mean chl fluorescence per cell was in heterotrophic cells, where in *C. vulgaris* cultures chl fluorescence per cell was lower than in photoautotrophic cells. Despite *C. reinhardtii* heterotrophic cultures growing minimally and decreasing in cell numbers after day 3, cells maintained higher chl fluorescence than in *C. vulgaris* heterotrophic cells (Figs. 6E, 6F).

The relationship between cell size estimates (FSC) and fluorescence per cell (FL-3) clearly differed between the three treatments (Fig. 7). During photoautotrophic growth of both species, the linear regression relationship showed very similar slopes across the two species and even closer similarity during exponential phase (Figs. 7A, 7B), but during mixotrophic and heterotrophic growth, the two species differed. Heterotrophic cultures showed tight linear regression relationships (highest $R^2$ values) between FSC and FL-3 but with lower FL-3 ranges (Figs. 7E, 7F). Mixotrophic cultures showed the most unconstrained FSC *vs.* FL-3 relationships with big differences between species and during exponential phase and over all time points and low $R^2$ values across all the time points (Figs. 7C, 7D).

## DISCUSSION

### Growth rates and biomass parameters

The two green algal species used in this study both showed trophic flexibility, growing well across a gradient of photoautotrophic-mixotrophic-heterotrophic conditions. The data support the hypothesis that growth rates differ with energy supply mode. Differences in growth rates and maximum biomass achievable may be important for selection of species and for choice of measurements. The data also supported the hypothesis that there are differences in growth rates based on different parameters used to assess biomass, and also

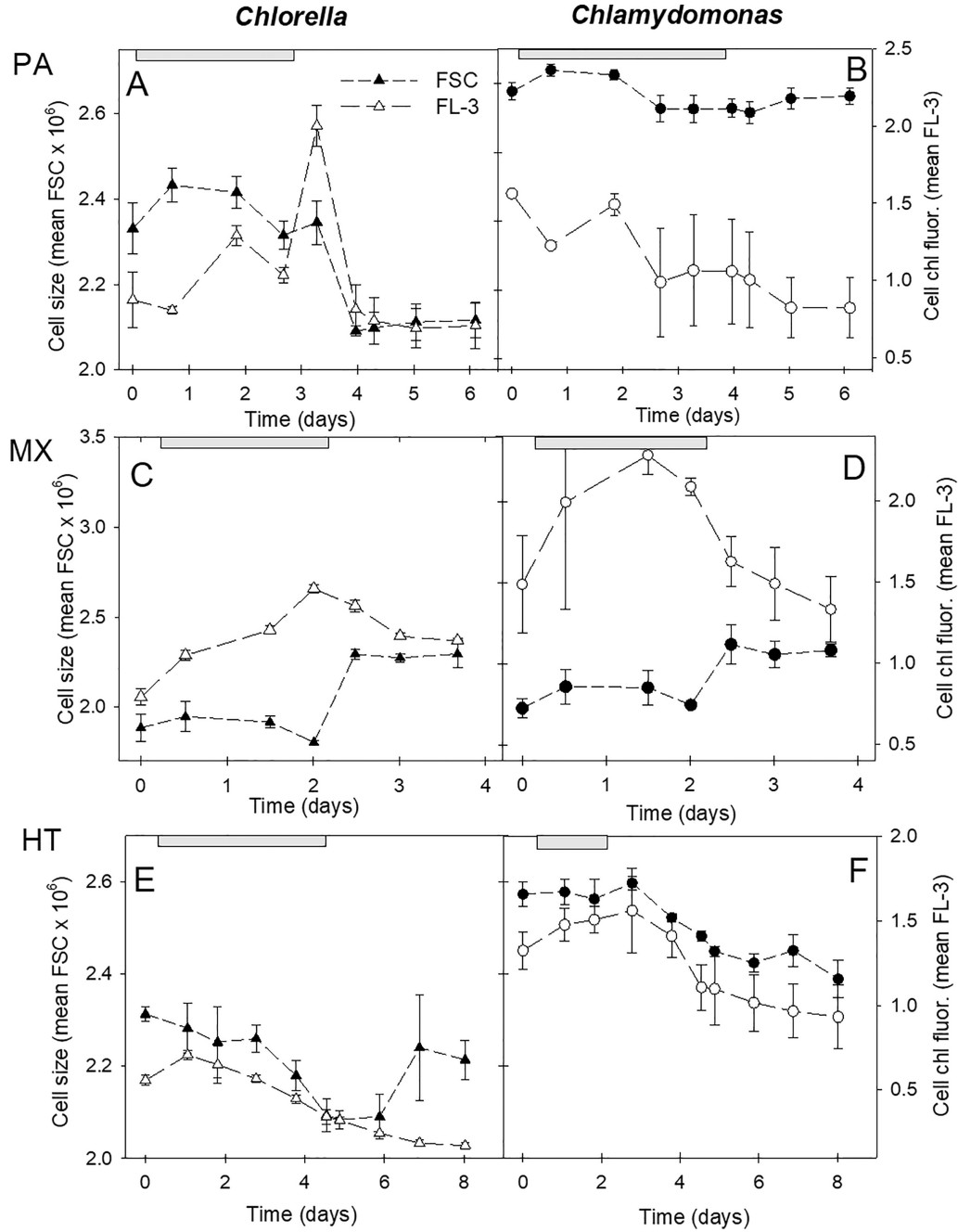

**Figure 6 Cell size and chl fluorescence per cell in cultures.** Changes in cell size and chl fluorescence per cell estimates over the culture growth period in the two algal species in photoautotrophic (A–B), mixotrophic (2 g/L acetate; C–D) and heterotrophic (E–F) growth conditions, for *Chlorella vulgaris* (A, C, E) and *Chlamydomonas reinhardtii* (B, D, F). Cell size estimates are based on the mean FSC-H for each sample (filled symbols, left scales) and chl fluorescence per cell (open symbols, right scales) is based on mean FL-3 values for each sample, both using the same flow cytometry gates determined for each species. Plots of the two species for each growth condition treatment are on the same scales. Exponential growth period is indicated with grey bars at top of plots. All points represent a mean of mean FSC-H or FL-3 values for three replicate cultures ± standard deviation. 

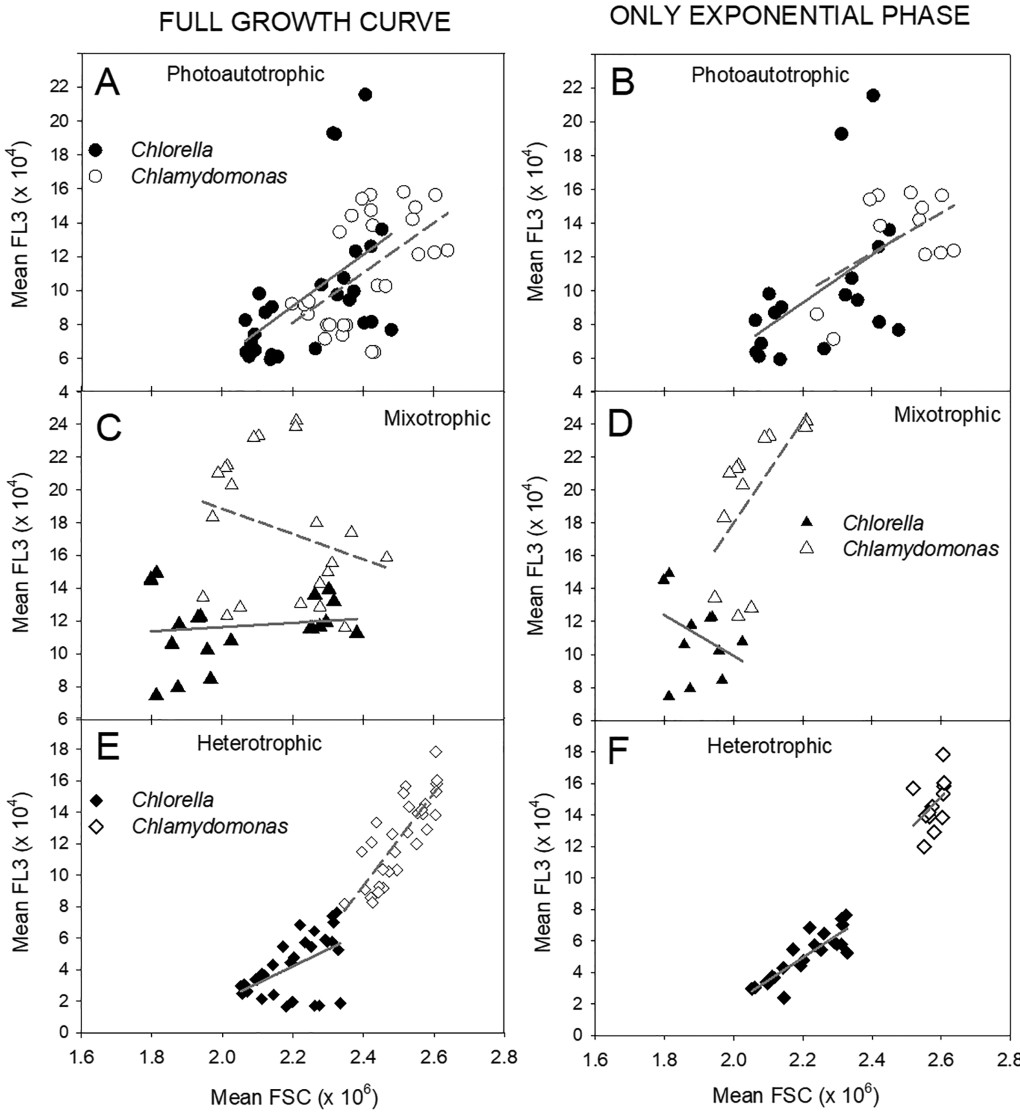

**Figure 7 Relationships between cell size and cell chl fluorescence.** Relationships between flow-cytometry estimates of cell size (mean FSC) and chl fluorescence per cell (mean FL-3) in *Chlorella vulgaris* (filled symbols) and *Chlamydomonas. reinhardtii* (open symbols) during photoautotrophic, mixotrophic (2 g/L acetate) or heterotrophic culture conditions. Points are mean FSC and FL-3 values from samples from three replicate cultures (mean values over time are shown in Fig. 6). A, C, E plots show all points over culture experiment and B, D, and F plots show points only during exponential growth phase. Lines for each species are fitted by linear regression. $R^2$ values for linear regressions: All points Exponential Growth Photoautotrophic: *Chlorella* – 0.263, *Chlamy* – 0.317, *Chlorella* – 0.247, *Chlamy* – 0.284; Mixotrophic: *Chlorella* – 0.0174, *Chlamy* – 0.0776, *Chlorella* – 0.134, *Chlamy* – 0.364; Heterotrophic: *Chlorella* – 0.284, *Chlamy* – 0.671 *Chlorella* – 0.747, *Chlamy* – 0.168.

that growth rates depend on energy supply. As previously noted (*Henley, 2019*), most studies do not report the exact time range used for calculations of growth rate, nor verify the exponential phase in batch culture. However, based on within-study comparisons reported, many studies suggest higher growth rates in mixotrophy relative to photoautotrophy or heterotrophy (Table 1). In *Chlamydomonas reinhardtii* and *Chlorella*
*vulgaris* cultures, any stimulation of growth rates in mixotrophic culture was mild, but also if a little acetate stimulated growth, then higher concentrations did not result in even higher growth, rather the highest acetate concentration suppressed growth rate and/or biomass yield. Most studies examining algal growth in different energy supply modes use just one concentration of organic C, but selected studies investigating concentration effects suggest a threshold organic C concentration, above which there is either no additional effect, or there is a decline in growth or biomass, observed in *Chlamydomonas, Scenedesmus* and *Isochrysis* (*Heifetz et al., 2000*; *Alkhamis & Qin, 2013*; *Bogaert et al., 2019*; *Cheng, Fan & Zheng, 2021*).

*C. vulgaris* was apparently less sensitive to exposure to acetate, with more even chl fluorescence-based growth rates across the gradient of photoautotrophic-mixotrophic-heterotrophic conditions, than *C. reinhardtii* which showed more variable growth rates with the highest in mixotrophic 2 g L$^{-1}$ acetate cultures. However, the exponential growth rate estimates also depended on the biomass parameter; growth rates based on OD were higher than for chl fluorescence, especially for heterotrophic conditions. OD is the most commonly used algal biomass parameter used (Table 1). Most studies report use of just one wavelength across 680–800 nm, without justification for the choice, although OD at 750 nm the most common (Table 1). Our findings of no differences between the three wavelengths suggests that results from studies using different OD wavelengths should still be relatively comparable.

Measuring OD at 680 nm is a logical choice to target a chl absorbance peak, but OD at 680 nm did not seem to have any closer relationship to chl fluorescence than 600 or 750 nm. A broad chl absorbance peak might include some overlap across 600 and 680 nm, although 750 nm would be out of chl absorbance range. OD is commonly used in large-scale cultures OD and/or with algal cultivation in wastewater (*Bogaert et al., 2019*; *Nirmalakhandan et al., 2019*), but in these conditions, cell and other debris could contribute absorbance at 750 nm, potentially overestimating algal cell biomass and therefore growth rate (*Chioccioli, Hankamer & Ross, 2014*). The higher biomass, and thus growth rate, based on OD than chl fluorescence suggests important distinctions in these measurements. Chl fluorescence per cell depends on cellular chl content but will also depend on cell size and vary with physiological status, including light acclimation and nutrient status (*Young & Beardall, 2003*).

We considered that the higher growth rates based on OD than chl fluorescence in heterotrophic culture could be due to contaminating non-photosynthetic biomass, which would contribute to OD but not chl fluorescence. Although bacteria are potentially beneficial to growth of algae in cultures (*Higgins & VanderGheynst, 2014*), to examine the possibility that bacteria were contributing to OD, cells were also counted using flow cytometry (FC) measurements allowing focus on the just the algal cell population, which are larger than bacteria. FC has been determined as a more sensitive parameter than OD for assessing growth of green algal species and can avoid any signal from cell debris (*Chioccioli, Hankamer & Ross, 2014*). In comparisons of growth rates based on changes in OD (680 nm), chl fluorescence, alongside microscopy (haemocytometer) cell counts and FC, OD clearly overestimated growth in heterotrophic cultures of *C. reinhardtii*, but not

convincingly so in *C. vulgaris* (Fig. 4). Another study of *Chlorella* also showed FC and microscopy counts more similar but differences with biomass and OD (*Chioccioli, Hankamer & Ross, 2014*). However, in contrast to this current study, *Chioccioli, Hankamer & Ross (2014)* also showed higher green algal growth rates based on FC than OD. In this study, more similar results between OD and the specific counting parameters (FC, microscopy counts) in *C. vulgaris*, suggests that OD is not overestimating growth because of bacterial contamination, but that high acetate and/or heterotrophic conditions produces algal cell biomass with suppressed or very low cell chl content. Overestimates using OD in *C. reinhardtii* cultures could be due to higher load of dead cells or debris. If algal cells are being grown for pigment production (*Benavente-Valdés et al., 2017*; *Kim et al., 2020*), then OD is a less suitable and direct pigment measurements or use of chl fluorescence would be more valuable for culture monitoring, and could be combined with other chl fluorescence parameters, including quantum yield to assess cell photosynthetic health (*Young & Beardall, 2003*; *Masojídek, Vonshak & Torzillo, 2010*; *Kamalanathan et al., 2017*).

## Other culture parameters

The two species showed similar growth rate ranges, maximum biomass achieved (OD or chl fluorescence) and time spent in exponential phase, so offer similar potential for application to bioreactor growth. The data supported the hypothesis of some apparent trade-offs between maximum growth rate and maximum biomass achieved and culture time spent in exponential phase. Based on OD, the highest *C. vulgaris* growth rates were in heterotrophic conditions but with a lower maximum biomass and a shorter exponential growth phase, and higher growth rates in heterotrophic *C. reinhardtii* cultures were accompanied by shorter time in exponential phase and lower biomass achieved. Similar trade-offs with lower growth rate but higher biomass accumulation was reported in *Chlamydomonas* grown mixotrophically with acetate (*Bogaert et al., 2019*) but in *Scenedesmus*, photoautotrophic cultures produced more biomass but at lower growth rate than in mixotrophic cultures (*Choi et al., 2019*). In another study, light-grown *Chlamydomonas* accumulated higher biomass in batch cultures with pulsed supply of acetate which prolonged the exponential growth phase (*Fields, Ostrand & Mayfield, 2018*) and higher biomass of *Chlorella* cells was achieved with acetate under mixotrophic conditions than photoautotrophy in batch cultures (*Karimian, Mahdavi & Gheshlaghi, 2022*). Higher biomass accumulation could be important in for yield in batch cultivation; in the two species examined, the exponential growth phase tended to be longer in light-grown than heterotrophic cultures. Maximum biomass achieved may be a critically important characteristic for economically viable commercial culture application (*Fu et al., 2012*; *Henley, 2019*). Longer time in exponential phase may allow more flexibility in harvesting timing for cultures, and/or provide longer window for cultures to be subsampled or growth stimulated by dilution or feeding (*Fields, Ostrand & Mayfield, 2018*).

Many studies comparing growth between energy acquisition modes report results of batch culture (Table 1), because they are cheaper and easier to maintain. Commercial

cultivation employs both batch culture and semi-continuous or continuous cultures, which maintain cells in exponential growth phase with continuous supply of nutrients and dilution to remove the inhibitory products which can accumulate in stationary phase (*Fernandes et al., 2015*; *Henley, 2019*). Hybrid batch or balanced chemostat bioreactors with organic C feeding or extra stages are also in development (*Fields, Ostrand & Mayfield, 2018*; *Kim et al., 2020*, *Abiusi et al., 2022*). It is unclear how batch culture parameters translate to continuous culture bioreactors or open-pond cultivation. Both species showed similar range of time in exponential phase, which may translate to relative stability in continuous cultures. However, the more similar growth rates across all conditions in *C. vulgaris* suggests that in commercial cultures supplemented with an organic source such as acetate under light limiting conditions, *C. vulgaris* growth may be more stable if organic C concentrations fluctuate.

## Chlorophyll and physiology

Culture supplementation with organic C can have profound effects on cell physiology, including cell pigment and protein content (Table 1) and macronutrient use (*Singh et al., 2014*; *Bogaert et al., 2019*; *Sim et al., 2019*; *Li et al., 2020*; *Wu et al., 2021*). The decrease in chl fluorescence with increasing concentrations of acetate, most evident in maximum chl fluorescence achieved (Figs. 3A, 3B), along with distinct patterns in cell chl fluorescence and cell size between the three energy acquisition modes (Fig. 7), suggests a clear effect of organic C use on chlorophyll synthesis, regulation of cell division and photosynthetic physiology. The marked decline in chl fluorescence above 4 g $L^{-1}$ acetate, and almost complete suppression in 10 g $L^{-1}$, suggests a threshold for suppression of investment in photosynthetic processes, and balancing growth demands with energy gain from heterotrophy. In *C. reinhardtii*, significantly lower chl fluorescence in all mixotrophic conditions suggests even low acetate concentration stimulates a down-regulation of photosynthesis, with a similar trend in *C. vulgaris*. Some previous studies show loss of chl with mixotrophic growth (Table 1) but others in which chl production was maintained even if photosynthetic capacity declined with increasing acetate availability (*Heifetz et al., 2000*). Mixotrophy can also suppress photosystem II or Rubisco activity (*Qiao, Wang & Zhang, 2009*; *Zili et al., 2017*) and marked transient effects of acetate on chlorophyll fluorescence indicate dynamic cell responses to acclimate carbon and energy metabolism (*Endo & Asada, 1996*). Other studies suggested maintenance of photosynthesis in mixotrophic growth may depend on species and/or organic substrate (*Kamalanathan et al., 2017*; *Cecchin et al., 2018*). Organic C supply can even support photosynthesis by reducing photoinhibition or improving $CO_2$ supply (*Xie et al., 2016*; *Curien et al., 2021*; *Gain et al., 2021*) and *Chlamydomonas* cells also show regulation of inorganic carbon acquisition in response to organic C supply, possibly by stimulating respiratory $CO_2$ production inside cells which can be used for photosynthesis (*Fett & Coleman, 1994*). However, some species need a light supply to be able to use glucose or to maximize growth benefit of organic C (*Chioccioli, Hankamer & Ross, 2014*; *Curien et al., 2021*). In contrast, diatoms may activate use of organic C when grown in the dark (*Tuchman et al., 2006*). Although chl content was profoundly affected in the green algae, similar maximum OD

values across the low-high acetate gradient suggests protein and cell wall synthesis was relatively less affected by exposure to acetate. Several studies suggest that mixotrophic growth conditions can also affect lipid production or accumulation in cells of some, but not all, algal and cyanobacterial species relative to PA and HT growth (Table 1) (*Ratha et al., 2013*; *Arora & Philippidis, 2021*). There are clearly complex interactions between photoautotrophic and heterotrophic processes, possibly specific to genotype, physiological conditions and organic substrate type.

## Cell chlorophyll and cell size

Changes in cell chl content, and chl fluorescence (*e.g.*, Fig. 2), were also be complicated by cell size changes, both over time in batch cultures, and in response to energy supply mode (Fig. 6), supporting the hypothesis. The relatively similar relationship in both species between FC cell size (FSC) and chl per cell (FL-3) between photoautotrophic and heterotrophic cultures, but differences under mixotrophy (Fig. 7), highlights the profound physiological acclimation associated with energy acquisition. The differences between these relationships during exponential phase *vs.* the full culture growth phases also indicate changes in physiological status of cells during batch culture growth phases. The relatively similar cell size *vs.* cell chl relationships both between species and between exponential phase and full culture growth in photoautotrophic cells suggests tighter constraints on cell chl content, as chl is critical for *all* energy harvesting. In heterotrophic cells, this relationship was also relatively tightly constrained, albeit with lower chl content. During mixotrophy (on 2 g L$^{-1}$ acetate), the much weaker relationship over the full culture phases suggests more dynamic and reduced physiological control over cell chl content, as energy harvesting can also rely on organic C use. In mixotrophic growth of *Chlamydomonas*, up to 50% of carbon can be derived from heterotrophy (*Heifetz et al., 2000*) reducing physiological investment in chl synthesis and photosynthetic energy harvesting (*Zili et al., 2017*). In heterotrophic culture, decline in cell chl over time (Figs. 6E, 6F) also indicates suppressed synthesis and/or decay of chl in the dark when energy metabolism is totally dependent on organic C uptake. Other studies have also reported loss of cell chl during mixotrophy (*Spijkerman, Lukas & Wacker, 2017*; *Li et al., 2020*) which can relate to reductions also in photosystem activity or photosynthetic capacity (*Heifetz et al., 2000*; *Zili et al., 2017*). In other studies, chl content or photosynthetic competence was not compromised during mixotrophy (*Laliberte & De la Noué, 1993*; *Cecchin et al., 2018*) and other pigments can be maintained despite organic C supplementation (*Kamalanathan et al., 2017*; *Kim et al., 2020*). The higher maintenance of chl per cell in mixotrophic and heterotrophic *C. reinhardtii* than in *C. vulgaris* (Fig. 6) might allow cells to re-acclimate to light more quickly in cultivation strategies combining different culture trophic modes (*Sim et al., 2019*; *Kim et al., 2020*).

Energy acquisition mode has also been shown to influence cell size, with larger cells shown by FC in cultures of *C. vulgaris* supplemented with glucose (*Chioccioli, Hankamer & Ross, 2014*). However, in this study, the smallest *C. vulgaris* and *C. reinhardtii* cells were observed in mixotrophic cultures during exponential phase (Fig. 6). *Chioccioli, Hankamer & Ross (2014)* also compared FC and microscopy, noting that OD and cell density were

more similar to FC counts during exponential growth phase, but cell counts diverged from OD values during stationary phase. In *Scenedesmus* cultures supplemented with molasses, the largest cell biovolumes were measured in mixotrophic, followed by heterotrophic cultures with the smallest cells in photoautotrophic cultures (*Kamalanathan et al., 2017*). In *Chlorella* and *Chlamydomonas* species, larger cells were reported in heterotrophic conditions (*Spijkerman, Lukas & Wacker, 2017*), and acetate feeding of *Chlamydomonas* resulted in larger cells than photoautotrophic growth (*Fields, Ostrand & Mayfield, 2018*). These observations were in contrast to our measurements. These inconsistent results across several studies just with green algae suggest that culture conditions, organic C source and concentration, and possibly genotype, can result in different responses of cell cycle regulation to energy acquisition mode.

## CONCLUSIONS AND RECOMMENDATIONS

These green algal species grew well across the gradient from photoautotrophic, a range of mixotrophic organic C concentrations, and heterotrophically. While organic C supplementation could support cell growth in these two commonly-used green algal species, light was required to maintain cell chl contents, cell size, as well as optimize growth rates and biomass yield using organic C. Estimates of algal growth depend on what parameter is used for calculating growth rates. Direct biomass measurements are valuable, but have been compared before (Table 1) and need large volumes and are labor-intensive. OD is easier and widely used but may overestimate growth of algal cells at higher organic C supply, or when water is turbid, for example in wastewater. Differences between rates based on OD and chl fluorescence may be important for mass cultivation for biotechnological applications. Alternative culture parameters need to be carefully compared for target species, particularly when cultures are supplemented with organic C. Examination of algal cultures with a single concentration of organic C source provides an incomplete picture of species response to mixotrophic conditions, which include differences in growth rates, cell size, chl per cell and maximum biomass at stationary phase of batch culture across a range of organic C concentrations. Furthermore, higher acetate concentration may inhibit, not promote growth rates; moderate organic C concentrations ($1–4$ g $L^{-1}$) with light may support highest growth rates and biomass yields.

## ACKNOWLEDGEMENTS

We acknowledge Lauren Simmons for help with flow cytometer measurements. Comments from anonymous reviewers were helpful.

### Funding

This study was supported by National Science Foundation grant CBET 1603196. The funders had no role in study design, data collection and analysis, decision to publish, or preparation of the manuscript.

## Grant Disclosures

The following grant information was disclosed by the authors:
National Science Foundation: CBET 1603196.

## Competing Interests

John A. Berges is an Academic Editor for PeerJ.

## Author Contributions

- Erica B. Young conceived and designed the experiments, analyzed the data, prepared figures and/or tables, authored or reviewed drafts of the article, and approved the final draft.
- Lindsay Reed performed the experiments, analyzed the data, prepared figures and/or tables, and approved the final draft.
- John A. Berges conceived and designed the experiments, analyzed the data, authored or reviewed drafts of the article, and approved the final draft.

## Data Availability

The raw data are available in the Supplemental File.

## Supplemental Information

Supplemental information for this article can be found online at http://dx.doi.org/10.7717/peerj.13776#supplemental-information.

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
