# Peer review of "Growth parameters and responses of green algae across a gradient of phototrophic, mixotrophic and heterotrophic conditions"

_PeerJ, doi:10.7717/peerj.13776_

## Round 0.1 · original submission · Minor Revisions

Noble Authors,

Three Experts expressed their opinion about your work. Everyone appreciated its high quality and innovative research. However, all of them had comments. Please kindly read these comments and respond to them.

With best regards,

Reviewer 1 ·

Basic reporting

The paper is very well written and the statistic is properly done. There is only one comment that needs to be taken into account. I have noticed in several places that the Authors use the wrong way of unit presentation such as g/L. This way is incorrect, especially since in other cases another way is applied.

Experimental design

It is correct.

Validity of the findings

In my opinion, this paper is a very important one as algal cultivation has practical application in CO2 capture and production of many important products. There are hundreds of possibilities to use the biomass. Concerning mixotrophy, it is not a totally new idea but the paper contains critical discussion and introduction that will be useful for many researchers and companies. The information is also important concerning the necessity to develop consortia of algae with other microbes to avoid problems with pathogens.

·

Basic reporting

1. Briefly explain what is the difference between PA,HT and MT cultivation system and pros and cons.
2. In the introductory part citation of latest references is required preferably last 5years including 2022.
3. some places authors used abbreviation, it is suggested to expand those abbreviations first and continue the abbreviation later in the text where ever required.

Experimental design

Line no: 159-162
Regarding acetate concentrations, authors mentioned they have used acetate ranges from 1-10g/l in the medium. is the same concentrations used for all trophic conditions(PA, HT, MT)? or they have used only 2g/l for HT condition?
it will give confusions to the readers, better clarification is required.
Table 1: update it with latest reports including 2022.

Validity of the findings

1.Figure captions and legend, reduce the word count, too lengthy captions are not looking nice for a figure. modification is required for all figure captions.
2. What is the reason behind variations in cell size during PA,HT & MT conditions with acetate, present discussion is not convinced. proper explanation is required
how chl is reducing or bleaching in HT, or chl molecule is changing to other molecule during heterotrophic system or other pigments are getting activated in HT, culture conditions. Include those point in line no 445-485.

Additional comments

Text is not fluid, check the grammatical and typological errors and English language need to be improved further.

Reviewer 3 ·

Basic reporting

The manuscript is well written, according with the jounral standards. It is well organized, structure is logic and coherent. Introduction fits to the main subject and justifies the problem undertaken. The cited literature is relevant to the scope of problem.
Graphical material clearly illustrates the experiment outcomes. I have found all of them necessary as they include results relevant to the hypotheses. Although the figures are readible, the overall quality (lines widths and fonts) needs improvement.

Experimental design

The manuscript is an interesting work on comparisons of cell culture parameters, and responses of cells over a gradient of mixotrophic, autotrophic and heterotrophic conditions for two species of green algae commonly used in biotechnology and for biofuels production. This is original primary research which falls in the scope of PeerJ journal. Reserch questions and problems have been clearly identified and justified.
Methods are described sufficiently to be replicated. Cultures and growth parameters of both algae species were examined according to the standards. Statistical methods (analysis of variance, mainly) seem enough to justify achieved differences. Resuls are properly documented.

Validity of the findings

The authors showed different parameters for estimating growth and biomass production under photoautotrophic, mixotrophic and heterotrophic growth conditions of Chlorella vulgaris and Chlamydomonas reinhardtii, which varied with a mode of energy supply. The results were a response to the formulated hypotheses. The results are statistically correct. The conclusions are relevant, supported by the results

Additional comments

It is a nice piece of work. Well organized and written.

---

## Round 0.2 · accepted · Accept

Dear Authors,
The reviewer stated that your work meets all the requirements and can be published in the PeerJ journal. My congratulations!

·

Basic reporting

Authors are nicely revised the manuscript, this manuscript can be accepted for publications

Experimental design

The authors are revised the manuscript well and modifications are done as per the comments

Validity of the findings

manuscript can be accepted for publications

Additional comments

manuscript can be accepted for publication